# Exercise versus Metformin to Improve Pregnancy Outcomes among Overweight Pregnant Women: A Systematic Review and Network Meta-Analysis

**DOI:** 10.3390/jcm10163490

**Published:** 2021-08-07

**Authors:** Carlos Pascual-Morena, Iván Cavero-Redondo, Celia Álvarez-Bueno, Maribel Lucerón-Lucas-Torres, Gema Sanabria-Martínez, Raquel Poyatos-León, Beatriz Rodríguez-Martín, Vicente Martínez-Vizcaíno

**Affiliations:** 1Health and Social Research Center, Universidad de Castilla—La Mancha, 16071 Cuenca, Spain; carlos.pascual@uclm.es (C.P.-M.); celia.alvarezbueno@uclm.es (C.Á.-B.); mariaisabel.luceron@uclm.es (M.L.-L.-T.); gema.sanabria@uclm.es (G.S.-M.); raquel.poyatos@uclm.es (R.P.-L.); beatriz.rmartin@uclm.es (B.R.-M.); vicente.martinez@uclm.es (V.M.-V.); 2Rehabilitation in Health Research Center (CIRES), Universidad de las Américas, Santiago 72819, Chile; 3Universidad Politécnica y Artística del Paraguay, Asunción 001518, Paraguay; 4Facultad de Ciencias de la Salud, Universidad Autónoma de Chile, Talca 3460000, Chile

**Keywords:** gestational diabetes mellitus, exercise, metformin, pregnancy, overweight, obesity, systematic review, network meta-analysis

## Abstract

Being overweight is associated with pregnancy-related disorders such as gestational diabetes mellitus (GDM), hypertensive disorders of pregnancy (HDP), and excessive maternal weight gain (MWG). Exercise and metformin reduce the risk of these disorders. This network meta-analysis (NMA) aims to compare the effect of metformin and different types of exercise (aerobic, resistance and combined) on the risk of GDM, HDP, and MWG among overweight/obese pregnant women. Medline, EMBASE, Web of Science and Cochrane Library were searched from inception to June 2021. Meta-analyses and NMAs were performed. Sixteen randomized controlled trials were included. In the NMA, aerobic exercise showed an effect on GDM (RR = 0.51, 95% CI = 0.26, 0.97), and metformin a reduction in MWG (MWG = −2.93 kg, 95% CI = −4.98, −0.87). No intervention showed any effect on the reduction of HDP. Our study suggests that aerobic exercise may have the greatest effect in reducing the risk of GDM, and perhaps, the MWG. Strategies should be developed to increase adherence to this type of intervention among overweight women without contraindications. Although metformin could reduce MWG, medicalization of pregnancy in healthy women is not justified with the present results. More research is needed on the effect of the intensity and frequency of exercise sessions and the length of interventions.

## 1. Introduction

Overweight and obesity are a major public health problem [1]. Obesity increases the risk of developing pregnancy complications, such as gestational diabetes mellitus (GDM), preeclampsia, or gestational hypertension [2]. GDM refers to any degree of carbohydrate intolerance with onset or first recognition during the second or third trimester of pregnancy [3], and has a prevalence about 17% [4,5]. GDM increases the risk of induced and caesarean deliveries and type 2 diabetes in women, as well as neonatal hypoglycemia, macrosomia, type 2 diabetes, obesity, and cardiovascular diseases in offspring [6,7,8,9,10].

It has been advocated that exercise could reduce the risk of GDM, excessive maternal weight gain (MWG) and hypertensive disorders of pregnancy (HDP) [11,12,13], an umbrella of disorders that includes the following clinical entities: pregnancy-induced hypertension (PIH), preeclampsia-eclampsia, chronic hypertension, and chronic hypertension with superimposed preeclampsia [14]. International recommendations suggest at least 30 min of moderate to vigorous physical activity, preferably aerobic or combined (resistance and aerobic exercise), per day, 3–4 times per week [15]. The effect of resistance exercise interventions on the health of pregnant women remains unclear [16]. However, the exercise recommendation for overweight/obese pregnant women remains a debatable issue [17].

Oral antidiabetics are another therapeutic strategy that has become widespread among women with polycystic ovary syndrome (PCOS) and obesity. Metformin reduces insulin resistance, endothelial dysfunction, and hyperglycemia, and it may prevent the development of HDP by reducing the secretion of soluble fms-like tyrosine kinase-1 (sFlt-1) [18,19,20,21,22]. However, a recent Cochrane review concluded that there is not enough evidence to support the use of metformin in women with obesity during pregnancy to improve maternal and infant outcomes [23].

A previous network meta-analysis (NMA) reported no effect of exercise or metformin on the risk of GDM and HDP (i.e., preeclampsia, PIH) among overweight/obese pregnant women, although it did show an effect of exercise on reducing MWG [24]. However, these two therapeutic options showed a clear trend towards a protective effect against GDM and HDP. Considering that the aforementioned NMA analyzed exercise as a single prescription, and that the results reported by the NMA were inconclusive but promising, we conducted this systematic review and NMA to comparatively synthesize data on the effect of metformin and different types of exercise interventions (i.e., aerobic, resistance, combined) on GDM, HDP and MWG among overweight/obese pregnant women.

## 2. Materials and Methods

This systematic review and NMA was conducted in accordance with the Preferred Reporting Items for Systematic Review incorporating NMA (PRISMA-NMA) guidelines and the Collaboration Handbook for Systematic Reviews of Interventions [25,26,27]. The study protocol was registered in PROSPERO (CDR:42019121715) and published elsewhere [28].

### 2.1. Search Strategy

Two reviewers (CP-M and IC-R) independently searched Medline, EMBASE, Web of Science and Cochrane Library databases from inception through June 2021. For unpublished trials, we searched through June 2021 clinicaltrials.gov, EudraCT and the grey literature, including Google Scholar, OPEN GRAY, Theseo and Networked digital library of theses and dissertations. We reviewed the reference list of articles included in this review and the list of previous systematic reviews. Relevant studies were identified using a combination of the following terms: (a) population; (b) interventions; and (c) clinical trials. The complete electronic search strategy is detailed in Appendix B.

### 2.2. Elegibility

Studies addressing the effect of exercise or metformin in pregnant women with overweight/obesity on the risk of GDM, HDP or MWG were included in the NMA. Inclusion criteria were as follows: (i) Type of study: Randomized controlled trials (RCTs), with no language restrictions; (ii) type of participants: Overweight or obese pregnant women; (iii) type of interventions: Metformin, structured exercise program; (iv) type of outcome assessment: Incidence of GDM, any HDP, and/or MWG (kg).

Exclusion criteria were as follows: (i) Single-arm pre-post studies, non-RCTs; (ii) studies whose target population was primarily women with PCOS, pregestational insulin resistance, or other diseases; (iii) dietary intervention as the primary co-intervention; (iv) nutraceutical interventions; or (v) unstructured physical activity/exercise intervention.

### 2.3. Data Extraction

Data from the included studies were extracted independently by two reviewers (CP-M and IC-R) according to the following predetermined information for each study: (i) Reference; (ii) study design; (iii) country; (iv) characteristics of intervention (type of intervention, length, intensity); (v) sample characteristics (type of population, sample size, mean age); and (vi) outcomes (risk of GDM, risk of HDP, MWG).

### 2.4. Categorization of Available Evidence

The classification of overweight or obesity was obtained from the classification made by the authors of the studies. When they did not report this classification, the internationally accepted values according to body mass index (BMI) were considered: overweight (BMI 25–30 kg/m^2^) and obese (BMI > 30 kg/m^2^) [1].

Exercise was defined as a subset of structured and repetitive physical activity with the objective of improving or maintaining physical fitness [29]. Exercise interventions were classified into three categories: (1) aerobic/endurance exercise; (2) strength/resistance exercise; and (3) combined exercise. Aerobic exercise included exercises such as swimming, cycling, jogging, running, and walking with the aim of increasing energy expenditure. Strength exercise included basic exercises with dumbbells and elastic bands aimed at increasing muscle strength. In combined interventions, aerobic exercise and strength exercises were alternated or combined.

The intensity of the exercise interventions was classified as light, light-moderate, moderate, moderate-vigorous, and vigorous, as reported by the authors. When intensity was not reported, criteria from the American College of Sports Medicine guidelines were used to estimate it [30,31,32].

### 2.5. Risk of Bias Assessment

Two researchers (CP-M and IC-R) independently conducted risk of bias assessment of included RCTs using the Cochrane Collaboration’s tool for assessing risk of bias (RoB2) [33]. The RoB2 evaluates risk of bias according to six domains: (1) Randomization process; (2) assignment to intervention; (3) adherence to intervention; (4) missing outcome data; (5) measurement of the outcome; and (6) selection of the reported result. Overall bias is considered “low risk of bias” if the study is classified as “low risk” in all domains, “some concerns” if there is at least one domain classified as “some concern”, and “high risk of bias” if there is at least one domain classified as “high risk” or several domains with “some concerns” that are considered critical to the validity of the results. Disagreements were resolved by discussion among the researchers, but if disagreements persisted, a third reviewer resolved the conflict (VM-V).

### 2.6. Grading of Quality of Evidence

The Grading of Recommendations, Assessment, Development and Evaluation (GRADE) tool was used to assess the quality of evidence and make recommendations [34,35]. Each outcome scored high, moderate, low, or very low evidence, according to study design, risk of bias, inconsistency, indirect evidence, imprecision, publication bias, large effect, possible confounding variables, and dose-response gradient.

### 2.7. Data Synthesis

The included RCTs were summarized qualitatively in an ad hoc table describing the types of direct and indirect comparisons. We conducted our NMAs in accordance with the PRISMA-NMA statement [26].

A network geometry graph was used to assess the robustness of the evidence in which the node size is proportional to the number of participants in the trial and the thickness of the continuous line connecting the nodes is proportional to the number of participants in the trials directly comparing the two treatments, and the dashed lines represent indirect comparisons [36,37].

We assessed consistency by testing whether the intervention effects estimated from direct comparisons were consistent with those estimated from indirect comparisons. Consequently, the Wald test was conducted, due to low statistical power, the side-splitting assessment was also used [38].

To compare the effects of different types of metformin/exercise interventions, a standard pairwise meta-analysis was performed for direct comparisons between interventions and controls/non-interventions. We used the DerSimonian–Laird method to perform all direct comparisons [39], and statistical heterogeneity was examined by calculating the I^2^ statistic, ranging from 0 to 100%. Heterogeneity was considered not important (<40%), moderate (30 to 60%), substantial (50 to 90%), or considerable (>75%) [27]. Additionally, the corresponding *p* values were considered. Finally, to determine the size and clinical relevance of the heterogeneity, the τ^2^ statistic was calculated. A τ^2^ estimate of 0.04 was interpreted as low, 0.14 as moderate and 0.40 as a substantial degree of clinical relevance of heterogeneity [40,41]. These results were displayed by creating both forest plots and a league table. For statistically significant results, we calculated the number needed to treat (NNT).

We assessed the principle of transitivity, which means checking that the synthesis of direct comparisons of two treatments has been conducted in similar studies on the most important clinical and methodological characteristics; thus, we assume that the populations included in these studies were similar in terms of the baseline distribution of effect modifiers, namely, baseline age and baseline BMI [42].

To identify superiority, the probability that each intervention was the most effective relative to the others was presented graphically using rankograms [36]. Additionally, the surface under the cumulative ranking (SUCRA) was estimated for each intervention. The SUCRA consists of assigning a numerical value between 0 and 1 to simplify the ranking of each intervention in the rankogram. The best intervention would obtain a SUCRA value close to 1, and the worst intervention a value close to 0 [37].

As a sensitivity analysis, we conducted a subgroup analysis including only pregnant women with obesity in which a standard meta-analysis and a NMA procedures were used for each outcome. Additionally, a sensitivity re-analysis under a Bayesian perspective was conducted. Furthermore, random effects meta-regression models showed that the age of the population included in the study did not influence the effect of exercise or metformin in GDM, HDP or MWG.

Finally, the small study effect and publication bias were assessed visually using a network funnel plot [43]. All analyses were conducted in Stata 15.0 (Stata, College Station, TX, USA).

### 2.8. Modifications to the Initial Protocol

In the study protocol [28], the target population was initially delimited to all pregnant women, including healthy women, overweight/obese women, or women with PCOS. In conducting this study, it was decided to include only women with overweight and obesity, and to conduct a subgroup study in women with obesity, to improve the transitivity principle. It was also planned to include any clinical trial, but it was decided to include only RCTs to increase the quality of the final analyses. Finally, the protocol included the performance of Bayesian NMAs. Subsequently, it was decided to conduct frequentist methods for NMAs, and a sensitivity analysis with Bayesian methods.

## 3. Results

Sixteen RCTs [44,45,46,47,48,49,50,51,52,53,54,55,56,57,58,59] were included in the analyses (Figure 1, Table 1, Appendix A), and 51 were excluded (Appendix A). Of the included studies, six included overweight pregnant women and 15 with obesity. The studies were conducted in 11 countries: Five in Europe (one in Ireland and one in Norway, two in the Netherlands, three in Spain and two in the United Kingdom), three in the Americas (one each in Canada and the United States and two in Brazil), one in Africa (Egypt), one in Asia (China) and one in Oceania (New Zealand). A total of 2903 pregnant women were included in the RCTs (412 in aerobic exercise, 925 in combined exercise, 40 in resistance exercise, and 1526 in metformin interventions). The interventions were generally performed before 20 weeks of gestation, generally at the end of the first trimester or at the beginning of the second trimester of gestation. The frequency of exercise sessions was two to five times per week, lasting 12 to 30 weeks. The dose of metformin was 1000 to 3000 mg daily, with a length of treatment of 20 to 25 weeks. Details of the interventions are described in Appendix A.

Fourteen studies focused on the prevention of GDM (three with aerobic exercise, six with combined exercise and five with metformin interventions), eleven on the prevention of HDP (three with aerobic exercise, four with combined exercise and five with metformin interventions), and twelve on the reduction of MWG (three with aerobic exercise, six with combined exercise, one with resistance exercise, and two with metformin interventions). Details of the GDM diagnostic criteria and the type of HDP included are described in Appendix A.

### 3.1. Gestational Diabetes Mellitus

Table 2A, Figure 2A, and Appendix A show the pairwise comparisons and NMA. Aerobic exercise showed a protective effect in the pairwise comparisons (upper diagonal) and in the NMA estimates (under diagonal) (RR = 0.59, 95% CI = 0.41, 0.85, and RR = 0.51, 95% CI = 0.26, 0.97, respectively). The NNT of aerobic exercise was 13 women to prevent one case.

### 3.2. Hypertensive Disorders of Pregnancy

Table 2B, Figure 2B, and Appendix A show that no intervention produced a significant effect on pairwise comparisons or NMA, although metformin almost reached statistical significance in the NMA (RR = 0.47, 95% CI = 0.22, 1.04).

### 3.3. Maternal Weight Gain

Table 2C, Figure 2C, and Appendix A show that aerobic exercise showed a protective effect in the pairwise comparisons, and metformin in NMA estimates (MWG = −1.91 kg, 95% CI = −2.74, −1.07, and MWG = −2.93 kg, 95% CI = −4.98, −0.87, respectively).

### 3.4. Risk of Bias

According to the Cochrane Collaboration tool for assessing risk of bias (RoB2), 10 out of 16 studies (62.5%) showed a high risk of bias for overall bias, and five (31.3%) showed some concerns. By domains, 6.3% of studies showed high risk for the randomization process; 18.8% and 56.3% showed high risk and some concerns, respectively, for assignment to an intervention; 43.8% and 18.8% showed high risk and some concerns, respectively, for adhering to an intervention; 18.8% and 18.8% showed high risk and some concerns, respectively, for missing outcome data; and 62.5% showed some concerns for measurement of the outcome. No significant risk of bias was detected for the selection of the reported results. The total risk of bias is detailed in Appendix A.

### 3.5. Grades of Recommendation, Assessment, Development, and Evaluation

According to the GRADE tool, aerobic exercise for the prevention of GDM, combined exercise for the prevention of HDP, and resistance exercise for the reduction MWG were scored as having low certainty, while the remaining interventions were scored as having very low certainty. All or almost all interventions were at serious or very serious risk of bias, inconsistency, and indirectness, and some interventions also had imprecision and publication bias. The full assessment is detailed in Appendix A.

### 3.6. Transitivity

There was no statistically significant difference in baseline age between the two interventions in any outcome. However, baseline BMI was higher in the metformin intervention compared with the exercise interventions. The transitivity study is detailed in Appendix A.

### 3.7. Probabilities

Aerobic exercise showed the highest probability of being the best intervention to prevent GDM (PrBest = 74.1%, SUCRA = 0.873), and metformin to prevent HDP and reduce MWG (PrBest = 64.3%, SUCRA = 0.850 and PrBest = 71.5%, SUCRA = 0.911, respectively) (Figure 3, Appendix A).

### 3.8. Subgroup Analysis among Pregnant Women with Obesity

Subgroup analysis showed no effect of any intervention for GDM and HDP, but metformin significantly reduced MWG (MWG = −2.88 kg, 95% CI = −5.37, −0.40) (Appendix A).

### 3.9. Sensitivity and Metarregresion Analyses

Analyses with Bayesian methods showed no statistically significant differences compared to frequentist analyses. Furthermore, random effects meta-regression showed no association between age and the effect of the interventions on the different outcomes.

### 3.10. Heterogeneity and Publication Bias

Aerobic and combined exercise showed no important heterogeneity and a low degree of clinical relevance of heterogeneity (I^2^ = 0.00% and τ^2^ = 0.00) for preventing GDM, HDP, and reducing MWG. Metformin and resistance exercise also showed no important heterogeneity and a low degree of clinical relevance of heterogeneity for preventing GDM and reducing MWG, respectively (I^2^ = 0.00% and τ^2^ = 0.00). Metformin showed considerable heterogeneity for preventing HDP and reducing MWG (I^2^ = 80.78% and I^2^ = 97.26%, respectively), and a substantial degree of clinical relevance of heterogeneity (τ^2^ = 0.76 and τ^2^ = 9.98, respectively) (Appendix A). Finally, there was evidence of publication bias only for MWG reduction (Appendix A).

## 4. Discussion

### 4.1. Main Findings

Our NMA shows that aerobic exercise reduces the risk of GDM by 49% in overweight/obese pregnant women, while metformin reduces MWG by 2.88–2.93 kg, depending on whether or not of women with overweight are included. Due to the effect size, the results also suggest that aerobic exercise may reduce MWG, and metformin the risk of HDP, which requires further research.

### 4.2. Interpretation

A previous meta-analysis reported a beneficial effect on GDM (RR = 0.71) [60] among overweight or obese women, but another meta-analysis found no effect [61]. In addition, the aforementioned NMA [24] also did not reach significance in its estimates and did not consider separately the effect of each type of exercise. Our data show that aerobic exercise is the recommended intervention to reduce the risk of GDM. The effect of exercise in preventing this pregnancy disorder could be due to the influence of exercise on improving glucose tolerance by increasing GLUT4 expression in muscle, skeletal muscle glycogen synthesis pathway activity, and TGF-β2 expression in fat tissue [62,63,64]. Moreover, aerobic exercise also improves pancreatic islet cell function, increases myonectin levels and decreases adipokine levels and oxidative stress [65,66]. Finally, it also increases caloric expenditure by increasing the number and size of mitochondria, carnitine transferase activity, and β-oxidation of fatty acids [63,65,66].

Previous meta-analyses showed an inconsistent effect of metformin on HDP. Thus, while one meta-analysis showed a protective effect (RR = 0.51) on the risk of preeclampsia [67], the others reported no effect [23,68,69]. Metformin had no effect on PIH [23,67,68,69]. Our results show no effect of metformin on HDP. However, it is difficult to rule out possible benefits of metformin on the risk of preeclampsia considering that metformin has been shown to improve endothelial dysfunction, poor vascularization, hypertension, and preeclampsia-related vasoconstriction by reducing soluble fms-like tyrosine kinase-1 and soluble endoglin levels; in addition, modulation of sirtuin 1 activity improves nitric oxide bioavailability through deacetylation of nitric oxide synthase. Moreover, closing the loop could also improve the activity of complex 1 of the mitochondrial electron transport chain, whose lower activity is associated with endothelial dysfunction [70,71].

Previous meta-analyses estimated [69,72] that metformin reduced MWG by 1.35 to 1.49 kg, whereas the previous NMA [24] found no effect. Moreover, regarding exercise, the previous meta-analysis and NMA showed reductions of 1.14 and 0.96 kg, respectively [24,60]. Our ranking of interventions shows that metformin is the most effective intervention to prevent excess MWG, followed by aerobic exercise. Aerobic exercise significantly reduces MWG, probably due to increased energy expenditure and fatty acid oxidation. Meanwhile, metformin reduces appetite by decreasing neuropeptide Y and Agouti-related protein and increases proopiomelanocortin and leptin and insulin sensitivity. In muscle, liver and adipose tissue, metformin increases fat oxidation and decreases hepatic glucose and fat synthesis by increasing AMPK activity [73,74,75].

Sensitivity analysis among pregnant women with obesity confirmed that metformin reduced the MWG. However, in women with obesity, aerobic exercise lost its effect on GDM. This was mainly due to the exclusion of the Wang C et al. study, as it had the one largest weight in the analyses, and the largest effect on GDM of all the aerobic exercise studies, which may be due, at least in part, to greater adherence and compliance of women to exercise sessions than in the other aerobic exercise studies.

Although our study has certain limitations, it is possible that increased adherence to exercise recommendations may reduce the risk of GDM and excess MWG in overweight/obese pregnant women. Although it is unclear what factors may increase adherence to physical activity recommendations, strategies to increase compliance such as increased health education in this group of women [76,77,78], should be implemented, and professionally supervised exercise, preferably aerobic exercise, and group exercise with other pregnant women could be recommended. Furthermore, metformin could reduce MWG, and perhaps, the risk of HDP. However, the potential benefits do not justify the medicalization of pregnancy, and the use of metformin should be carefully assessed in each woman, considering the specific benefits and risks.

### 4.3. Limitations

Some limitations must be acknowledged. First, the scarcity of studies could influence the publication bias analyzes, the assessment of transitivity requirement, and the statistical power of the effect estimates, especially in aerobic exercise, which included only three studies. Second, due to the scarcity of studies, the effect of length, frequency, intensity, or other covariates could not be analyzed, being a substantial source of heterogeneity among interventions. Third, the diagnostic criteria of GDM varied among the included studies. Fourth, the outcome HDP as a single clinical entity, including PIH and pre-eclampsia, two related entities but which might respond differently to each intervention. Fifth, we found no studies on resistance exercise for GDM and HDP that met our inclusion criteria. Sixth, moderate to high risk of bias was found in most studies, with especially important being whether the high risk was due to adhering to intervention or missing outcome data domains. Seventh, transitivity analysis showed a difference in BMI baseline in the exercise and metformin interventions. Eighth, although the meta-regression analyses did not find an association between age and effect, due to the low number of studies this association cannot be ruled out.

## 5. Conclusions

Active pregnancy is currently recommended to prevent some pregnancy-related complications among overweight/obese pregnant women, including the risk of GDM, HDP, and excessive MWG. One of the limitations of previous studies was to determine which type of exercise is most effective in preventing pregnancy-related complications. Our study suggests that aerobic exercise is the most effective in reducing the risk of GDM and, perhaps, MWG. Metformin showed an effect in reducing MWG, although the lack of effect on other outcomes, the low quality of evidence and, especially, the risks associated with its administration, do not justify the medicalization of pregnancy in healthy pregnant women with obesity. Finally, due to the limitations of our study, further research is needed on the effect of intensity and weekly frequency of exercise sessions and length of interventions, as well as further RCTs in this population group to increase the statistical power of the results.

## Figures and Tables

**Figure 1 jcm-10-03490-f001:**
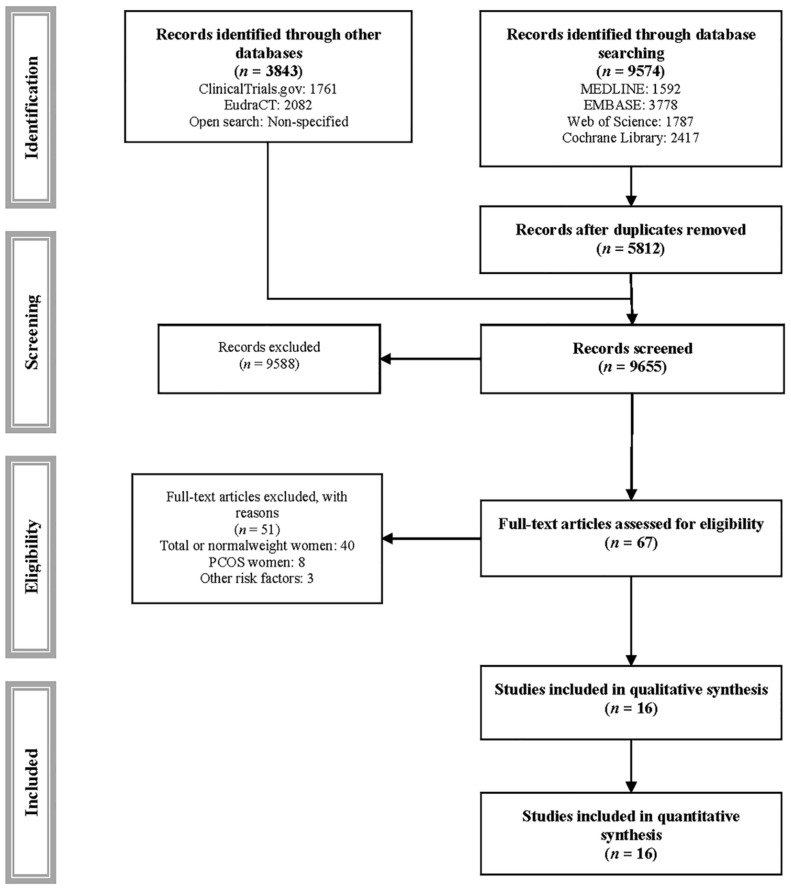
PRISMA flowchart of study selection.

**Figure 2 jcm-10-03490-f002:**
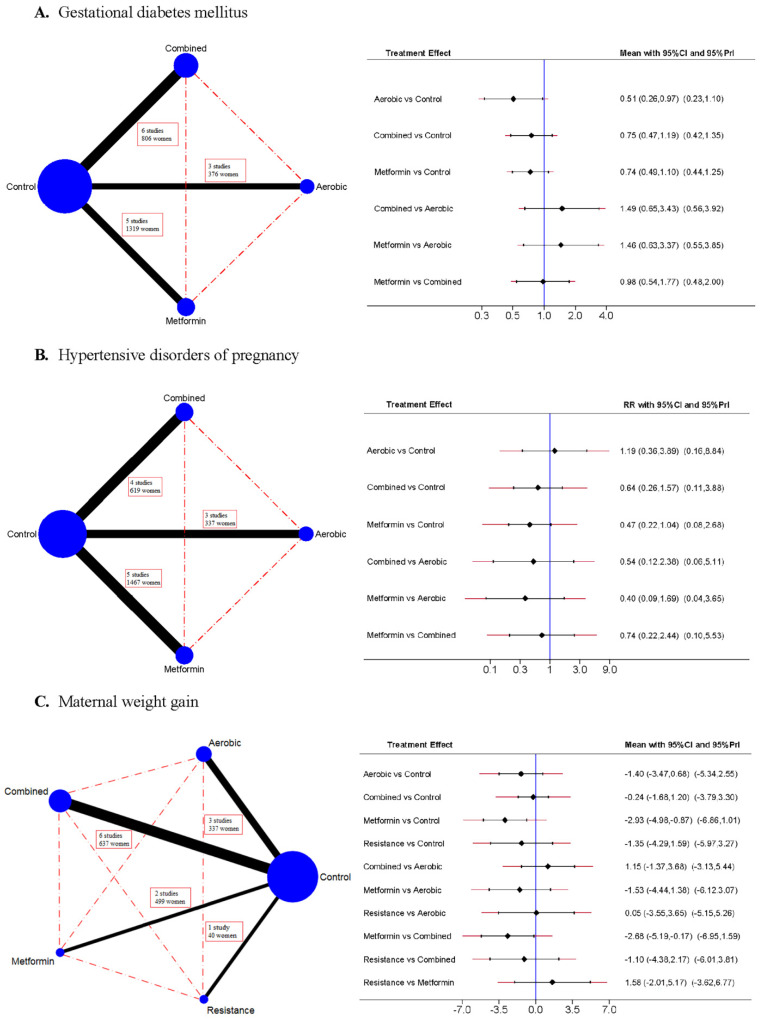
Network mappings and network meta-analyses estimates for gestational diabetes mellitus (**A**), hypertensive disorders of pregnancy (**B**) and maternal weight gain (**C**). The network mappings (**left**) represent the number of participants in each intervention arm, and direct and indirect comparisons. The forest plot/interval plot (**right**) represent the effect size estimates in the network meta-analysis. The risk ratio (RR) was estimated for dichotomous variables, and the main difference (Mean) for continuous variables, with their 95% confidence intervals.

**Figure 3 jcm-10-03490-f003:**
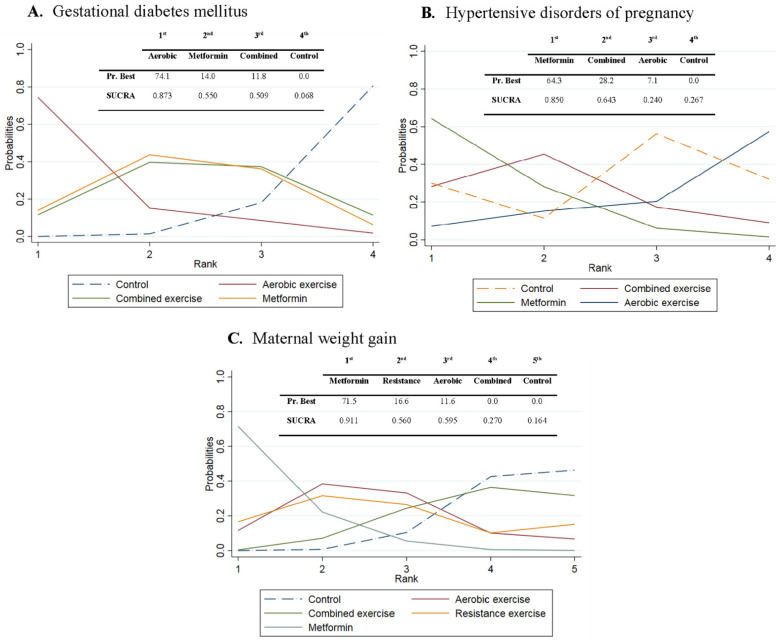
Relative rankings of treatments for gestational diabetes mellitus (**A**), hypertensive disorders of pregnancy (**B**) and maternal weight gain (**C**). Graphs of the rankograms for each outcome. In each graph, the data of each intervention were detailed in a table for the probability of being the best intervention (Pr. Best) and its SUCRA.

**Table 1 jcm-10-03490-t001:** Characteristics of included trials.

Reference	Design	Country	Intervention	Target Women	Sample			Age		Outcome		
					T	I	C	I	C	GDM	HDP	MWG
**Kong K et al. (2014)-1 [44]**	RCT	United States	Aerobic exercise	Overweight	19	9	9	26.2 ± 2.6	27.3 ± 3.6	**✓**	**✓**	**✓**
**Kong K et al. (2014)-2 [44]**	RCT	United States	Aerobic exercise	Obese	18	9	10	28.6 ± 5.3	25.7 ± 4.0	**✓**	**✓**	**✓**
**Seneviratne SN et al. (2016) [45]**	RCT	New Zealand	Aerobic exercise	Obese	75	38	37	NA	NA	**✓**	**✓**	**✓**
**Wang C et al. (2017) [46]**	RCT	China	Aerobic exercise	Overweight	300	150	150	32.1 ± 4.6	32.5 ± 4.9	**✓**	**✓**	**✓**
**Barakat R et al. (2016)-1 [47]**	RCT	Spain	Combined exercise	Overweight	164	90	78	NA	NA	**✓**	**✓**	-
**Barakat R et al. (2016)-2 [47]**	RCT	Spain	Combined exercise	Obese	54	25	29	NA	NA	**✓**	**✓**	-
**Bisson M et al. (2015) [48]**	RCT	Canada	Combined exercise	Obese	50	25	25	30.5 ± 3.7	31.0 ± 4.0	**✓**	**✓**	**✓**
**Daly N et al. (2017) [49]**	RCT	Ireland	Combined exercise	Obese	88	44	44	30.0 ± 5.1	29.4 ± 4.8	**✓**	-	**✓**
**Garnæs KK et al. (2016) [50]**	RCT	Norway	Combined exercise	Obese	91	46	45	31.3 ± 3.8	31.4 ± 4.7	**✓**	**✓**	**✓**
**Nascimento SL et al. (2011) [51]**	RCT	Brazil	Combined exercise	Overweight and Obese	82	40	42	29.7 ± 6.8	30.9 ± 5.9	-	-	**✓**
**Oostdam N et al. (2012) [52]**	RCT	Netherlands	Combined exercise	Obese	121	62	59	30.8 ± 5.2	30.1 ± 4.5	**✓**	-	**✓**
**Ruiz JR et al. (2013) [53]**	RCT	Spain	Combined exercise	Overweight and Obese	275	146	129	NA	NA	**✓**	**✓**	**✓**
**Barakat R et al. (2009)-1 [54]**	RCT	Spain	Resistance exercise	Overweight	28	14	14	NA	NA	-	-	**✓**
**Barakat R et al. (2009)-2 [54]**	RCT	Spain	Resistance exercise	Obese	12	9	3	NA	NA	-	-	**✓**
**Abd El Fattah EA et al. (2016) [55]**	RCT	Egypt	Metformin	Obese	200	100	100	26.9 ± 5.2	26.2 ± 5.5	**✓**	**✓**	**✓**
**Brink HS et al. (2018) [56]**	RCT	Netherlands	Metformin	Obese	49	24	25	29.3 ± 5.2	30.7 ± 5.2	**✓**	**✓**	-
**Chiswick C et al. (2015) [57]**	RCT	UK	Metformin	Obese	449	226	223	28.7 ± 5.8	28.9 ± 5.1	**✓**	**✓**	**✓**
**Nascimento IB et al. (2020) [58]**	RCT	Brazil	Metformin	Obese	378	189	189	28.6 ± 6.2	29.6 ± 6.1	**✓**	**✓**	-
**Syngelaki A et al. (2016) [59]**	RCT	UK	Metformin	Obese	450	225	225	32.9	30.8	**✓**	**✓**	-

T: Total; I: Intervention; C: Control; GDM: Gestational diabetes mellitus; HDP: Hypertensive disorders of pregnancy; MWG: Maternal weight gain.

**Table 2 jcm-10-03490-t002:** Results for direct pairwise comparisons and network meta-analysis.

A. Gestational Diabetes Mellitus					
	Control	Aerobic	Combined	Resistance	Metformin
**Control**		0.59 * (0.41, 0.85)	0.91 (0.67, 1.22)		0.78 (0.59, 1.02)
**Aerobic**	0.51 * (0.26, 0.97)		-		-
**Combined**	0.75 (0.47, 1.19)	1.49 (0.65, 3.43)			-
**Resistance**					
**Metformin**	0.74 (0.49, 1.10)	1.46 (0.63, 3.37)	0.98 (0.54, 1.77)		
**B.** Hypertensive disorders of pregnancy					
**Control**		0.95 (0.56, 1.62)	0.65 (0.36, 1.14)		0.48 (0.19, 1.22)
**Aerobic**	1.19 (0.36, 3.89)		-		-
**Combined**	0.64 (0.26, 1.57)	0.54 (0.12, 2.38)			-
**Resistance**					
**Metformin**	0.47 (0.22, 1.04)	0.40 (0.09, 1.69)	0.74 (0.22, 2.44)		
**C.** Maternal weight gain					
**Control**		−1.91 * (−2.74, −1.07)	−0.31 (−1.06, 0.44)	−1.35 (−3.57, 0.88)	−2.82 (−7.26, 1.62)
**Aerobic**	−1.40 (−3.47, 0.68)		-	-	-
**Combined**	−0.24 (−1.68, 1.20)	1.15 (−1.37, 3.68)		-	-
**Resistance**	−1.35 (−4.29, 1.59)	0.05 (−3.55, 3.65)	−1.10 (−4.38, 2.17)		-
**Metformin**	−2.93 * (−4.98, −0.87)	−1.53 (−4.44, 1.38)	−2.68 * (−5.19, −0.17)	−1.58 (−5.17, 2.01)	

Light gray indicates lack of direct and indirect comparisons. Dark gray separates direct from indirect comparisons. Gestational diabetes mellitus and hypertensive disorders of pregnancy were measured as risk ratio (95% CI). Maternal weight gain was measured as mean difference (95% CI). Upper diagonal: Standard meta-analysis. Lower diagonal: Network meta-analysis estimates. * Statistical significance.

## Data Availability

The datasets generated and analyzed are available from the corresponding author.

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
