# Peer review of "Exercise versus Metformin to Improve Pregnancy Outcomes among Overweight Pregnant Women: A Systematic Review and Network Meta-Analysis"

_jcm, 2021, doi:10.3390/jcm10163490_

Round 1

Reviewer 1 Report

The manuscript submitted by Dr. Pascual-Morena et al. presents a systematic review and network meta-analysis comparing the effects of various types of exercise and metformin on pregnancy outcomes in overweight/obese pregnant women. The paper is well-written, using an elegant and accurate English. The Introduction section clearly and concisely presents the rationale for the study, and formerly published data in this field. The current study is precisely designed, with clear inclusion/exclusion criteria. The Authors paid attention to the risk of bias assessment and grading of quality of evidence. Several bioinformatic tools were applied to ensure proper data synthesis. The results are clearly presented in a series of tables and graphs. They indicate that aerobic exercise reduces the risk of GDM in pregnant overweight/obese women, while metformin decreases maternal weight gain during pregnancy. Furthermore, they also suggest that aerobic exercise could reduce maternal weight gain, and metformin could decrease the frequency of the hypertensive disorders of pregnancy. These data are of particular interest, especially that metformin is still not approved for routine usage in pregnancy in many countries. The results are then thoroughly discussed with skillful reference to biochemical and metabolic basis and to previously published data. Moreover, the limitations of the study are also explicitly acknowledged. Finally, the conclusions are precise and strongly supported by the results of the meta-analysis performed.

Overall, this is a scientifically sound and clinically important paper.

Author Response

Authors

We really appreciate the review, time spent, and the comments made by the reviewer.

Reviewer 2 Report

This Network Meta-analysis compared the relative impact of metformin and exercise on obese pregnant women on the development of GDM, pregnancy related hypertension and weight gain. The authors found found 16 RCTs fulfilling their inclusion criteria. Aerobic exercise impacted the development of GDM and Metformin impacted weight gain. No intervention reduced the development of pregnancy related hypertension.:

1/ It would be useful to know when during the course of pregnancy that these interventions were implemented. An intervention which may have had an impact in the first trimester could have been too late if implemented in the third trimester, Related to this it would have been useful to also have assessed studies that have implemented these strategies pre-conception.

2/ The authors recognize that there is significant heterogeneity in the interventions eg frequency of exercise sessions was 2 to 5 times per week, lasting 12 to 30 weeks and in addition 10 of 16 studies (62.5%) showed a high risk of bias and 5 (31.3%) showed some concerns which does limit the value of this analysis

3/ It may be that the exercise and metformin interventions could have had a more discernable impact in those overweight women at greater risk of GDM/ hypertension/ weight gain. Could the authors provide any insights as to the impact of the average age of the study population on the outcome?

4/ The grammar should be tidied up eg 

This Network Meta-analysis compared the relative impact of metformin and exercise on obese pregnant women on the development of GDM, pregnancy related hypertension and weight gain. The authors found found 16 RCTs fulfilling their inclusion criteria. Aerobic exercise impacted the development of GDM and Metformin impacted weight gain. No intervention reduced the development of pregnancy related hypertension.:

1/ It would be useful to know when during the course of pregnancy that these interventions were implemented. An intervention which may have had an impact in the first trimester could have been too late if implemented in the third trimester, Related to this it would have been useful to also have assessed studies that have implemented these strategies pre-conception.

2/ The authors recognize that there is significant heterogeneity in the interventions eg frequency of exercise sessions was 2 to 5 times per week, lasting 12 to 30 weeks and in addition 10 of 16 studies (62.5%) showed a high risk of bias and 5 (31.3%) showed some concerns which does limit the value of this analysis

3/ It may be that the exercise and metformin interventions could have had a more discernable impact in those overweight women at greater risk of GDM/ hypertension/ weight gain. Could the authors provide any insights as to the impact of the average age of the study population on the outcome?

4/ The grammar should be tidied up eg title

Author Response

  • It would be useful to know when during the course of pregnancy that these interventions were implemented. An intervention which may have had an impact in the first trimester could have been too late if implemented in the third trimester, Related to this it would have been useful to also have assessed studies that have implemented these strategies pre-conception.

Authors

Thank you for the reviewer's comment. As suggested, we have included this information in the results section as follows:

“The interventions were generally performed before 20 weeks of gestation, generally at the end of the first trimester or at the beginning of the second trimester of gestation.”

  • The authors recognize that there is significant heterogeneity in the interventions eg frequency of exercise sessions was 2 to 5 times per week, lasting 12 to 30 weeks and in addition 10 of 16 studies (62.5%) showed a high risk of bias and 5 (31.3%) showed some concerns which does limit the value of this analysis

Authors

The reviewer’s comment seems judicious. We have specified that it is a source of heterogeneity among interventions in limitations section as follows:

“Second, due to the scarcity of studies, the effect of length, frequency, intensity or other covariates could not be analyzed, being a substantial source of heterogeneity among interventions.”

  • It may be that the exercise and metformin interventions could have had a more discernable impact in those overweight women at greater risk of GDM/ hypertension/ weight gain. Could the authors provide any insights as to the impact of the average age of the study population on the outcome?

Authors

Thank you for the reviewer’s comment. As suggested, we have performed a meta-regression using the mean age of women included in the studies. The following information has been included in the data synthesis, results, and limitations sections:

Data synthesis

“Furthermore, random effects meta-regression models showed that the age of the population included in the study did not influence the effect of exercise or metformin in GDM, HDP or MWG.”

Results

“Furthermore, random effects meta-regression showed no association between age and the effect of the interventions on the different outcomes.”

Limitations

“Eighth, although the meta-regression analyses did not find an association between age and effect, due to the low number of studies this association cannot be ruled out.”

  • The grammar should be tidied up eg title

Authors

Thank you for the reviewer's comment. A full review of the manuscript has been performed by a native English-speaker of our research center.